# Prognostic Value of the Controlling Nutritional Status (CONUT) Score in Patients at Dialysis Initiation

**DOI:** 10.3390/nu14112317

**Published:** 2022-05-31

**Authors:** Kimiaki Takagi, Hiroshi Takahashi, Tomomi Miura, Kasumi Yamagiwa, Kota Kawase, Yuka Muramatsu-Maekawa, Takuya Koie, Masashi Mizuno

**Affiliations:** 1Department of Urology, Daiyukai Daiichi Hospital, 1-6-12, Hagoromo, Ichinomiya 491-0025, Japan; yuka_mae11@yahoo.co.jp; 2Department of Urology, Gifu University Graduate School of Medicine, 1-1 Yanagido, Gifu 501-1193, Japan; stnf55@gifu-u.ac.jp (K.K.); goodwin@gifu-u.ac.jp (T.K.); 3Department of Nephrology, Fujita Health University School of Medicine, Toyoake 470-1192, Japan; hirotaka@fujita-hu.ac.jp; 4Department of Nutrition, Daiyukai Daiichi Hospital, 1-6-12, Hagoromo, Ichinomiya 491-0025, Japan; 2109007.bunri@gmail.com (T.M.); kasumi.xxxxx@gmail.com (K.Y.); 5Renal Replacement Therapy, Division of Nephrology, Nagoya University Graduate School of Medicine, 65 Tsurumai, Showa-ku, Nagoya 166-8550, Japan; mmizu@med.nagoya-u.ac.jp

**Keywords:** protein-energy wasting, CONUT, chronic kidney disease, dialysis, mortality, nutritional assessment

## Abstract

Protein-energy wasting (PEW) is common in patients with chronic kidney disease (CKD), and affects their prognosis. The Controlling Nutritional Status (CONUT) score is a nutritional screening tool calculated using only blood test data. This study aimed to investigate the prognostic value of CONUT score in patients just initiating dialysis. A total of 311 CKD patients who stably initiated dialysis were enrolled. Only 27 (8.7%) patients were classified as having normal nutritional status. The CONUT score was also independently correlated with elevated C-reactive protein levels (β = 0.485, *p* < 0.0001). During the median follow-up of 37 months, 100 patients (32.2%) died. The CONUT score was an independent predictor of all-cause mortality (adjusted hazard ratio 1.13, 95% confidence interval 1.04–1.22, *p* < 0.0024). As model discrimination, the addition of the CONUT score to a prediction model based on established risk factors significantly improved net reclassification improvement (0.285, *p* = 0.028) and integrated discrimination improvement (0.025, *p* = 0.023). The CONUT score might be a simplified surrogate marker of the PEW with clinical utility and could predict all-cause mortality, in addition to improving the predictability in CKD patients just initiating dialysis. The CONUT score also could predict infectious-disease mortality.

## 1. Introduction

Protein-energy wasting (PEW), a state of decreased body protein mass and energy fuel, is frequently observed in chronic kidney disease (CKD) patients [1]. The PEW, a CKD-specific nutritional disorder, affects various clinical outcomes [2,3,4]. 

Various nutritional screening tools such as Subjective Global Assessment (SGA), Mini Nutritional Assessment (MNA), Geriatric Nutrition Risk Index (GNRI), or Creatinine (Cr) index, have been widely developed [5,6,7,8]. Among them, the Malnutrition Inflammation Score (MIS) and the PEW criteria by the International Society of Renal Nutrition and Metabolism (ISRNM), including more details such as serum chemistry, measurement of body mass, muscle mass, or dietary intake, have been established as a golden standard to assess the PEW [9,10]. 

On the other hand, the Controlling Nutritional Status (CONUT) score, a simplified nutritional assessment tool using only laboratory test data, has been developed [11]. The CONUT score is also known to predict the prognosis of patients with cancer and acute heart failure [12,13,14,15]. However, there are limited studies that have investigated the association between CONUT score and the prognosis of patients with CKD [16]. This study aimed to verify whether the CONUT score is useful for predicting the prognosis in CKD patients just initiating dialysis therapy.

## 2. Materials and Methods

### 2.1. Study Population and Design

This was a retrospective, single-center, observational cohort study. CKD patients who initiated hemodialysis (HD) or peritoneal dialysis (PD) were enrolled in this study between 1 January 2012 and 31 December 2020, at Daiyukai Daiichi Hospital. Patients were followed up from the initiation of dialysis until 31 December 2021, or the last clinical encounter. Patients who initiated dialysis for acute kidney injury (AKI) were excluded.

The requirement for informed consent was waived due to the anonymity of the data. Instead of providing individual consent, we notified the subjects about this study by posting documents at prominent places in the hospital and guaranteed refusal opportunities from participating in the research for patients (the opt-out option). The study was approved by the Ethics Committee of Shakai Iryo Hojin Daiyukai (approval number: 2021-012) and was conducted in accordance with the Declaration of Helsinki. 

### 2.2. Data Collection and Measurements

The CONUT score is shown in Appendix A [11]. The CONUT score was calculated based on serum albumin concentration, total lymphocyte count, and total cholesterol level. Patients were classified into four nutritional states according to the score depending on their blood test results—namely, normal, light, moderate, and severe. Clinical and laboratory data, including serum levels of albumin, total lymphocyte count, total cholesterol, hemoglobin, and C-reactive protein (CRP), which were measured at dialysis initiation, were obtained from the individual medical records. Body mass index (BMI) was calculated using “dry weight” determined after the dialysis session. We also assessed the cardiothoracic ratio (CTR) before dialysis initiation. Cardiovascular diseases (CVDs) included myocardial infarction, arrhythmia, heart failure, stroke, or sudden death. We also surveyed the number of patients who were referred to a nephrologist less than three months before dialysis initiation as a definition of late referral. We diagnosed comorbidities, such as diabetes mellitus, hypertension, and dyslipidemia, according to the patients’ medical history and medication status. Smoking was defined either as a current smoker or one who had smoked in the past but quit smoking.

### 2.3. Follow-Up Study

Follow-up was censored in December 2021. Patients who were lost to follow-up for reasons such as transfer from Daiyukai Daiichi Hospital were censored at the date of the last contact. Patients who were alive on 31 December 2021 were censored for the overall survival analysis. Overall survival was calculated from the date of dialysis initiation to the date of death from any cause. The primary endpoint was all-cause death and the secondary endpoints were death from CVDs and infectious diseases.

### 2.4. Statistical Analysis

Continuous variables with a normal distribution are expressed as mean ± standard deviation (SD), and asymmetrically distributed data are presented as the median and interquartile range (IQR). Differences among the CONUT score groups were evaluated using the chi-square test for categorical variables, ANOVA for normally distributed continuous variables, and Kruskal–Wallis test for asymmetrically distributed continuous variables. To determine the factors that correlated with the CONUT score, multivariate regression analyses were performed including baseline variables (*p* < 0.05) in the univariate analysis. Survival was estimated using the Kaplan–Meier method, and the differences in survival were compared using the log-rank test. We used Cox proportional hazard regression models to examine the predictors of all-cause, CVDs, and infectious disease mortality. Significant baseline variables (at *p* < 0.05) in the univariate analysis were included in the multivariable models. 

We also calculated the C-index, net reclassification improvement (NRI), and integrated discrimination improvement (IDI) to assess whether the accuracy of predicting mortality would improve after the addition of the CONUT score into a baseline model that included established risk factors; that is, significant baseline variables (at *p* < 0.05) in the univariate analysis. In addition, we compared the predictability of the predicting model with established risk factors plus CONUT with those plus GNRI [7]. The threshold for significance was set at *p* < 0.05. All statistical analyses were conducted using JMP version 14.3.0. (SAS Inc., Cary, NC, USA).

## 3. Results

### 3.1. Characteristics of the Study Population

In total, 328 consecutive patients were eligible for the study. After applying the exclusion criteria, 311 patients (HD, *n* = 273; PD, *n* = 38) were enrolled in this study. Baseline patient characteristics are shown in Table 1. Based on the CONUT score, the number of patients in the four groups: normal, light, moderate, and severe, were 27 (8.7%), 134 (43.1%), 120 (38.6%), and 30 (9.6%), respectively. The baseline albumin, total cholesterol, and lymphocyte counts were 3.3 ± 0.7 g/dL, 170 ± 52 mg/dL, and 1023 (777–1377) /μL, respectively. In the high CONUT score group, the patients’ age (*p* = 0.0116), the percentage with CVD history (*p* = 0.0264) and CRP levels were higher (*p* < 0.0001). On the other hand, in the group with low CONUT scores, PD was selected more frequently as the dialysis method (*p* = 0.0015), and hemoglobin levels was higher (*p* < 0.0001). Multivariate regression analysis showed that the CONUT score was independently correlated with age (β = 0.025, *p* = 0.03), diabetes mellitus (β = 0.695, *p* = 0.009), CRP (β = 0.485, *p* < 0.0001), and hemoglobin (β = −0.578, *p* < 0.0001) (Table 2).

### 3.2. Prognostic Value of the CONUT Score

During the follow-up period (median 37 months, IQR 17–63 months), 3 patients underwent kidney transplantation, and 78 patients were transferred to another dialysis institution. They were censored at the date of the last contact. A total of 100 patients (32.2%) died, including 39 deaths (39%) due to CVDs and 33 deaths (33%) due to infectious diseases (Table 3). 

In the univariate Cox proportional hazards analysis, CONUT score (as a continuous variable), age, dyslipidemia, BMI, history of CVDs, and CRP were significant predictors of all-cause mortality (Appendix A). CONUT score (as a continuous variable) was an independent predictor of all-cause mortality after adjustment for other confounders (HR 1.13, 95% CI 1.04–1.22, *p* = 0.0024). Kaplan–Meier mortality rates for 7-year were 27.2%, 42.7%, 56.0%, and 79.8% in the normal, light, moderate, and severe CONUT groups, respectively (Figure 1A). The results of univariate and multivariate analyses to identify the predictive value of the CONUT score for mortality by cause of death are shown in Table 4. The severe CONUT score group had a 5.47-fold higher all-cause mortality risk, compared with the normal group (HR 5.47, 95% CI 1.19–25.2, *p* = 0.029). 

The C-index for all-cause mortality was calculated for the model discrimination. The C-index tended to improve (0.712, *p* = 0.086). However, the NRI and IDI for all-cause mortality significantly improved after the CONUT score was added to the baseline model, along with established risk factors (0.285 and 0.025, *p* = 0.028 and *p* = 0.023, respectively). The C-index, NRI, and IDI for infectious disease mortality significantly improved (0.711, 0.486 and 0.035, *p* = 0.035, *p* = 0.007 and *p* = 0.002, respectively; Table 5). 

Regarding the comparison of each predictive model, the model plus CONUT has an almost comparable predictive ability for all-cause mortality compared with the model plus GNRI (C-index; 0.702 vs. 0.690, *p* = 0.28), whereas the C-index of the model plus CONUT for infectious disease mortality tended to improve (0.711 vs. 0.664, *p* = 0.084; Table 6).

### 3.3. Sub-Analysis by Cause of Death

CONUT score (as a continuous variable) was an independent predictor of infectious disease mortality after adjustment for other confounders (HR 1.28, 95% CI 1.11–1.47, *p* = 0.0006). Kaplan–Meier mortality rates for infectious diseases for 7 years were 0%, 12.0%, 31.2%, and 34.7% in the normal, light, moderate, and severe CONUT groups, respectively (Figure 1B). However, estimating HR for infectious disease mortality between the CONUT score groups was not appropriate because no infectious death occurred in the normal group. On the other hand, the CONUT score was not a significant predictor of CVD mortality (HR 1.05, 95% CI 0.92–1.2, *p* = 0.4388). Kaplan–Meier mortality rates for CVDs for 7 years were 5.3%, 25.2%, 22.5%, and 17.3% in the normal, light, moderate, and severe CONUT groups, respectively (Figure 1C). There were no significant differences in CVD mortality risk among the CONUT score groups (*p* = 0.75).

## 4. Discussion

The results showed that the CONUT score strongly predicted all-cause mortality while also improving the predictive accuracy of mortality with increasing NRI and IDI in CKD patients who had just initiated dialysis therapy. Furthermore, the CONUT score was a significant prognostic factor for infectious diseases death but not for CVDs death.

Considering nutritional disorders, PEW, a state of nutritional and metabolic derangement characterized by the simultaneous loss of systemic body protein and energy stores, is an important issue in patients with CKD. PEW is caused by hypercatabolic states, uremic toxins, malnutrition, and inflammation from systemic conditions such as CKD. Albumin, a component of the CONUT score, has traditionally been used as an indicator of malnutrition or a predictor of mortality in patients with CKD [17]. Inflammatory states are the main factor causing low albumin levels [18,19]. Inflammation negatively affects albumin synthesis [20]. A previous study showed that the serum CRP level, an inflammatory marker, increases continuously in PEW [21]. In the present study, CRP level was an independent risk factor for all-cause and CVD mortality, and the higher nutritional risk CONUT score was significantly associated with a higher CRP level. Therefore, the CONUT score is suggested as a useful screening tool for PEW in CKD patients just initiating dialysis therapy. 

Although dialysis itself can cause nutritional disorders, CKD patients are expected to already have PEW at the time of dialysis initiation. According to our study data, less than 10% of patients were classified as having normal CONUT scores. This seems to indicate that the PEW was existing at dialysis initiation. Thus, it is important to screen the nutritional status of CKD patients at the time of dialysis initiation. 

CVDs and infectious diseases account for a large proportion of the deaths in dialysis patients. In Japan, the percentage of patients who died from CVDs, including heart failure, cerebrovascular disease, and myocardial infarction, was 33.1%, and that of infectious diseases was 21.3% [22]. The proportion of deaths in our cohort is similar to that in Japan. 

According to past literature, undernutrition may reduce lymphocyte maturation and circulating lymphocyte counts [23]. A decrease in the total lymphocyte count can be an indicator of malnutrition. 

Nutritional disorders disrupt immune health and compromise resistance to and recovery from infections [24]. In patients with CKD, the function of the immune system is affected by multiple factors, including oxidative stress and inflammation, accumulation of uremic toxins, increased apoptosis of immune cells, and disturbed renal metabolic effects. Immune dysfunction in patients with CKD resulting from these factors is responsible for the increased propensity for infectious diseases [25]. A study of patients hospitalized for acute decompensated heart failure showed that a high CONUT score was associated with a higher risk of in-hospital mortality and infections [26]. Although the CONUT score is associated with various clinical outcomes, a low lymphocyte count in the CONUT score could directly indicate vulnerability to infectious diseases. Our study showed a robust association, especially with mortality from infectious diseases in dialysis patients. It is not a sensitive measure because it is easily affected by coexisting diseases and stress. None of these variables are specific to nutritional status and may be influenced by other factors. When using the CONUT score as a nutritional screening tool, it is necessary to consider the patients’ background and comorbidities. 

On the other hand, the CONUT score was not associated with CVD mortality. Although dyslipidemia is an established risk factor for CVDs, high cholesterol levels are a low-risk factor for malnutrition in CONUT. In dialysis patients, there is a concept of “reverse epidemiology”, whereby obesity, hypercholesterolemia, and hypertension appear to be protective features and are associated with a greater survival [27,28]. This may explain why CONUT scores did not predict CVD mortality in this study. 

Lastly, numerous nutritional assessment tools, including MIS and PEW criteria established by ISRNM as a golden standard to assess the PEW, have been widely developed and clinically used. However, these tools may be somewhat troublesome when used in daily clinical practice because of needing measurement of body mass, muscle mass, or dietary intake. In comparison, the CONUT score is a simplified tool using only laboratory tests; thus, it may have high clinical utility in daily practice.

The CONUT score was initially shown to be as useful as the SGA and Full Nutritional Assessment, as a nutritional status screening tool, for inpatients [11]. MIS and the creatinine index are widely used as nutritional indicators for patients with CKD [9,29]. Each of these indicators has been shown to be associated with clinical outcomes, such as all-cause mortality and the onset of CVDs [9,30], but the evaluations of these tools are complicated. SGA [5] and Mini Nutritional Assessment (MNA) [31,32], which are screening tools that evaluate nutritional status by combining interviews, medical history, and physical measurements, are also complex indicators. As subjective items are also included in these tools, errors may occur depending on the individual patient and examiner. 

In addition, comparing predictability with GNRI, which is one of the widely used nutritional indicators calculated only with objective data so as to CONUT, it seemed to be almost comparable for all-cause mortality. On the other hand, CONUT tended to be superior in predicting infectious disease mortality. Thus, CONUT possibly might be a nutritional indicator specifically for infectious disease mortality. This finding needs to be confirmed in a large cohort in the future.

This study has several limitations. First, it was a retrospective study with relatively small sample size. Second, in the analysis, we did not consider the modality of dialysis. It was difficult to obtain an exact index for standardized dialysis for each patient because some patients changed their dialysis modality during the follow-up period. Third, we only checked the patients’ data at dialysis initiation, and the impact of nutritional status changes on prognosis was not analyzed after dialysis initiation.

## 5. Conclusions

The CONUT score might be a simplified surrogate marker of the PEW with clinical utility and one that can predict all-cause mortality in addition to improving predictability in CKD patients just initiating dialysis. The CONUT score can also predict infectious disease mortality.

## Figures and Tables

**Figure 1 nutrients-14-02317-f001:**
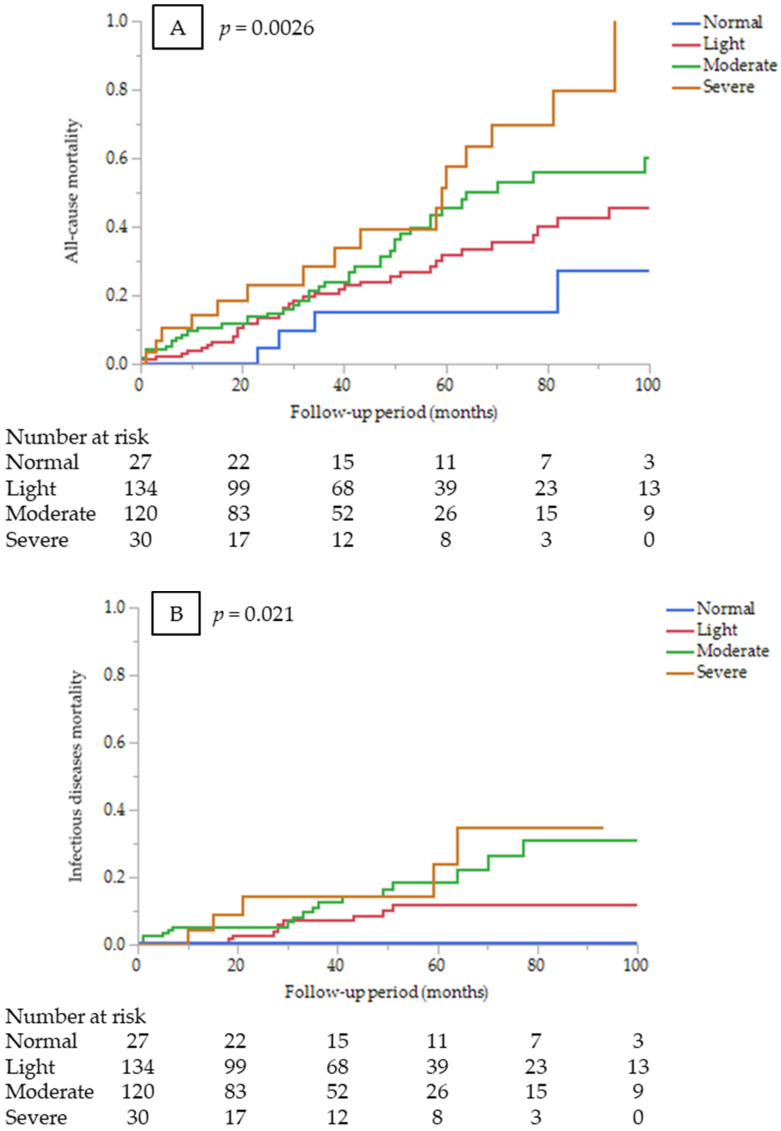
Kaplan–Meier survival curves comparing among the CONUT score groups at dialysis initiation (**A**) for all-cause mortality (log-rank test, *p* = 0.0026); (**B**) infectious disease mortality (log-rank test, *p* = 0.0211); and (**C**) CVD mortality (log-rank test, *p* = 0.3789).

**Table 1 nutrients-14-02317-t001:** Patient characteristics according to CONUT score groups.

Variables	All *n* = 311	CONUT Score Groups				*p* Value
		Normal *n* = 27	Light *n* = 134	Moderate *n* = 120	Severe *n* = 30	
Age (years)	69 ± 12	63 ± 14	69 ± 12	70 ± 13	73 ± 11	0.0116
Sex, male (%)	226 (73)	16 (59)	99 (74)	89 (74)	22 (73)	0.4755
Smoking, yes (%)	148 (48)	13 (48)	58 (43)	61 (51)	16 (53)	0.5923
History of CVDs (%)	134 (43)	6 (22)	55 (41)	55 (46)	18 (60)	0.0264
Late referral (%)	33 (11)	2 (7)	12 (9)	15 (13)	4 (13)	0.729
Diabetes mellitus (%)	170 (55)	15 (56)	64 (48)	71 (59)	20 (67)	0.148
Hypertension (%)	275 (88)	25 (93)	122 (91)	104 (87)	24 (80)	0.3097
Dyslipidemia (%)	110 (35)	11 (41)	44 (33)	43 (36)	12 (40)	0.8007
BMI	21.9 ± 3.9	23.4 ± 4.8	21.9 ± 3.8	21.7 ± 3.8	20.8 ± 3.3	0.091
CTR	53.3 (48.6–58.8)	49.3 (45.6–52.9)	53.5 (49–59.3)	53.3 (48.3–58.4)	56.2 (53.4–60)	0.0038
Etiology of ESRD (%)						0.1625
Diabetic nephropathy	165 (53)	14 (52)	62 (46)	70 (58)	19 (63)	
Non-diabetic nephropathy	146 (47)	13 (48)	72 (54)	50 (42)	11 (37)	
Dialysis modality, PD (%)	38 (12)	4 (15)	25 (19)	9 (8)	0 (0)	0.0015
Albumin (g/dL)	3.3 ± 0.7	3.8 ± 0.3	3.7 ± 0.4	2.9 ± 0.5	2.2 ± 0.4	<0.0001
Total cholesterol (mg/dL)	170 ± 52	194 ± 39	174 ± 40	168 ± 63	133 ± 41	<0.0001
Lymphocyte count (/μL)	1023 (777–1377)	1758 (1233–4585)	1100 (360–2350)	881 (160–4311)	783 (172–1942)	<0.0001
CRP (mg/dL)	0.2 (0.07–0.6)	0.07 (0.02–0.33)	0.12 (0.05–0.4)	0.35 (0.09–0.92)	0.37 (0.16–1.53)	<0.0001
Hemoglobin (g/dL)	9.2 ± 1.5	9.7 ± 1.3	9.6 ± 1.4	8.9 ± 1.5	8.3 ± 1.3	<0.0001

Abbreviations: CONUT, Controlling Nutritional Status; BMI, body mass index; CVDs, cardiovascular diseases; CTR, cardi thoracic ratio; ESRD, end-stage renal disease; PD, peritoneal dialysis; CRP, C-reactive protein.

**Table 2 nutrients-14-02317-t002:** Relationship between CONUT score and baseline characteristics at univariate and multivariate regression analyses.

	Univariate		Multivariate	
	β	*p*-Value	β	*p*-Value
Age	0.1714	0.0024	0.0252	0.0305
Sex, male	0.0382	0.5025		
Smoking	0.0638	0.2619		
History of CVD	0.1552	0.0061	0.5112	0.0540
Late referral	0.0697	0.2204		
Diabetes mellitus	0.1380	0.0149	0.6953	0.0092
Hypertension	−0.1155	0.0419	−0.3758	0.3583
Dyslipidemia	0.0306	0.5909		
BMI	−0.0890	0.1172	−0.0465	0.1910
CTR	0.1400	0.0136	−0.0130	0.4887
Log CRP	0.3463	<0.0001	0.4851	<0.0001
Hemoglobin	−0.3790	<0.0001	−0.5776	<0.0001

Abbreviations: CONUT, Controlling Nutritional Status; BMI, body mass index; CVD, cardiovascular disease; CTR, thoracic ratio; CRP, C-reactive protein.

**Table 3 nutrients-14-02317-t003:** Distribution of deceased dialysis patients by cause of death.

Cardiovascular diseases	21 (21%)
Cerebrovascular diseases	8 (8)
Sudden death	10 (10)
Infectious diseases	33 (33)
Malignancy	12 (12)
Cachexia	4 (4)
Others	7 (7)
Undetermined	5 (5)

**Table 4 nutrients-14-02317-t004:** Predictive value for all-cause mortality, CVD mortality, and infectious disease mortality.

Variables	Univariate		Multivariate *	
	HR (95% CI)	*p* Value	HR (95% CI)	*p* Value
**All-cause mortality**				
CONUT score (continuous)	1.18 (1.09–1.27)	<0.0001	1.13 (1.04–1.22)	0.0024
CONUT score groups (vs. normal)		0.0031 **		0.019 **
Light	2.3 (0.82–6.45)	0.11	2.75 (0.64–11.8)	0.17
Moderate	3.21 (1.15–8.97)	0.026	3.93 (0.92–16.8)	0.065
Severe	5.38 (1.79–16.1)	0.0027	5.47 (1.19–25.2)	0.029
**CVD mortality**				
CONUT score (continuous)	1.12 (0.99–1.27)	0.059	1.05 (0.92–1.2)	0.4388
CONUT score groups (vs. normal)		0.25 **		0.75 **
Light	4.2 (0.56–31.6)	0.16	2.1 (0.26–17.2)	0.49
Moderate	4.94 (0.65–37.3)	0.12	2.68 (0.33–21.9)	0.36
Severe	5.37 (0.6–48.2)	0.13	1.8 (0.17–19.6)	0.63
**Infectious disease mortality**				
CONUT score (continuous)	1.30 (1.15–1.49)	<0.0001	1.28 (1.11–1.47)	0.0006
CONUT score groups (vs. normal) ^¶^		0.0058 **		0.032 **

* Adjusted for age, dyslipidemia, BMI, CVD, and CRP as variables with *p* < 0.05 by univariate analysis **; *p* for trend ^¶^; estimating HR was not appropriate because no infectious death occurred in normal group. Abbreviations: CI, confidence interval; CONUT, Controlling Nutritional Status; CVDs, cardiovascular diseases; HR, hazard ratio.

**Table 5 nutrients-14-02317-t005:** Discrimination of the predicting models for all-cause mortality and infectious disease mortality using C-index, net reclassification improvement (NRI), and integrated discrimination improvement (IDI).

Variables	C-Index	*p* Value	NRI	*p* Value	IDI	*P* Value
**All-cause Mortality**						
Established risk factors *	0.676	Reference	Reference		Reference	
+CONUT	0.712	0.086	0.285	0.0278	0.025	0.023
**Infectious diseases Mortality**						
Established risk factors *	0.63	Reference	Reference		Reference	
+CONUT	0.711	0.035	0.486	0.007	0.035	0.002

* Established risk factors included age, dyslipidemia, adjusted BMI, previous CVD, and CRP level as significant baseline variables at *p* < 0.05 in the univariate analysis. Abbreviations: CONUT, Controlling Nutritional Status; BMI, body mass index; CVDs, cardiovascular diseases; CRP, C-reactive protein.

**Table 6 nutrients-14-02317-t006:** Discrimination of each predicting model with CONUT or GNRI for all-cause mortality and infectious disease mortality using C-index, net reclassification improvement (NRI), and integrated discrimination improvement (IDI).

Variables	C-Index	*p* Value	NRI	*p* Value	IDI	*p* Value
**All-cause Mortality**						
Established risk factors * +GNRI	0.69	Reference	Reference		Reference	
Established risk factors * +CONUT	0.702	0.282	0.051	0.339	0.009	0.061
**Infectious diseases Mortality**						
Established risk factors * +GNRI	0.664	Reference	Reference		Reference	
Established risk factors * +CONUT	0.711	0.084	0.063	0.367	0.004	0.295

* Established risk factors included age, dyslipidemia, previous CVD, and CRP level as significant baseline variables at *p* < 0.05 in the univariate analysis. Abbreviations: CONUT, Controlling Nutritional Status; GNRI, Geriatric Nutritional Risk Index; CVDs, cardiovascular diseases; CRP, C-reactive protein.

## Data Availability

The data presented in this study are available upon request from the corresponding author. The data are not publicly available for privacy and ethical reasons.

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
