# Peer review of "Prognostic Value of the Controlling Nutritional Status (CONUT) Score in Patients at Dialysis Initiation"

_nutrients, 2022, doi:10.3390/nu14112317_

Round 1
Reviewer 1 Report
In the Introduction the authors said that no studies have investigated the association between CONUT score and the prognosis of patients with CKD which is incorrect. In just one minutes search two articles were found: DOI: 10.1016/j.clnu.2019.11.018 and DOI: 10.1016/j.clnu.2019.11.018 therefore authors should perform comprehensive literature search to put the topic of the research into perspective.
Author Response
In the Introduction the authors said that no studies have investigated the association between CONUT score and the prognosis of patients with CKD which is incorrect. In just one minutes search two articles were found: DOI: 10.1016/j.clnu.2019.11.018 and DOI: 10.1016/j.clnu.2019.11.018 therefore authors should perform comprehensive literature search to put the topic of the research into perspective.
Response: Thank you very much for your appropriate comment. As reviewer’s comment, “no studies have investigated the association between CONUT score and the prognosis of patients with CKD” is incorrect. We changed the sentence to “there are limited studies which have investigated the association between hemodialysis patients and CONUT” in the revised manuscript. We are deeply sorry for our carelessness.

Reviewer 2 Report
Authors reported the prognostic value of CONUT score on top of classical risk factors in patients with CKD. Although the report have some interesting findings, current version of the manuscript does not reach the standards of publication in the Nutrients.
* Many previous reports have demonstrated the usefulness of nutritional markers/scores added to classical risk factors to predict patient's prognosis. Thus, it is not so novel just to report another score which can be a prognostic marker. In addition, GNRI, MNA-SF are also very easy to calculate with variables which is readily available. Thus, authors should compare other scores (such as GNRI, MNA-SF) to show the superiority to use CONUT in predicting prognosis.
* Other way to find the superiority of CONUT may be its predicting ability for infectious prognosis. In this case, still author needs to compare other nutritional scores to show the superiority of CONUT.
Author Response
* Many previous reports have demonstrated the usefulness of nutritional markers/scores added to classical risk factors to predict patient's prognosis. Thus, it is not so novel just to report another score which can be a prognostic marker. In addition, GNRI, MNA-SF are also very easy to calculate with variables which is readily available. Thus, authors should compare other scores (such as GNRI, MNA-SF) to show the superiority to use CONUT in predicting prognosis.
* Other way to find the superiority of CONUT may be its predicting ability for infectious prognosis. In this case, still author needs to compare other nutritional scores to show the superiority of CONUT.
Response: Thank you very much for your important comments. We fully agree your point-out. According to reviewer’s suggestion, we added the results of comparison of the predictability between CONUT model and GNRI model as Table 6 in the revised manuscript. CONUT was not superior to GNRI's ability to predict the prognosis of all-cause mortality (C-index; 0.702 vs. 0.690, p=0.28). However, for infectious death, there was a tendency for improvement of predictability for CONUT model vs, GNRI model (C-index; 0.711 vs. 0.664, P = 0.084). Thus, we think the CONUT possibly might be a nutritional indicator specifically for infectious diseases mortality. In addition, to emphasize this, we added the improvement of predictability for infectious death in Table 5 in the revised manuscript.
Unfortunately, we could not calculate MNA-SF score because the subjective items used to score the MNA-SF was not included in the data collected in the present study.
We thanks for your kind understanding.
